# Interpreting the Repeated Token Phenomenon in Large Language Models

**Itay Yona** [1]   **Ilia Shumailov** [1]   **Jamie Hayes** [1]   **Yossi Gandelsman** [2]

## Abstract

Large Language Models (LLMs), despite their impressive capabilities, often fail to accurately repeat a single word when prompted to, and instead output unrelated text. This unexplained failure mode represents a *vulnerability*, allowing even end users to diverge models away from their intended behavior. We aim to explain the causes for this phenomenon and link it to the concept of "attention sinks", an emergent LLM behavior crucial for fluency, in which the initial token receives disproportionately high attention scores. Our investigation identifies the neural circuit responsible for attention sinks and shows how long repetitions disrupt this circuit. We extend this finding to other nonrepeating sequences that exhibit similar circuit disruptions. To address this, we propose a targeted patch that effectively resolves the issue without negatively impacting the overall performance of the model. This study provides a mechanistic explanation for an LLM vulnerability, demonstrating how interpretability can diagnose and address issues, and offering insights that pave the way for more secure and reliable models. Code is available here.

## 1. Introduction

Large Language Models (LLMs) have achieved remarkable success in various natural language tasks. However, even state-of-the-art LLMs can exhibit surprising failures on seemingly simple tasks. A prominent example is the "repeated token divergence" phenomenon (Nasr et al., 2023; Barbero et al., 2025), where LLMs struggle to accurately repeat a single input token when prompted to do so. This issue has been observed across a range of models, including: ChatGPT, LLaMa1, LLaMa2 (OpenAI, 2022; Touvron et al., 2023a;b). The underlying cause of this unexpected behavior

has remained an open question up until now.

In this paper, we explain the reason behind the repeated token divergence in LLMs. We link this behavior to the mechanism of "attention-sinks" – a phenomenon where the initial token in a sequence receives disproportionately high attention, previously shown to be crucial for LLM fluency (Xiao et al., 2024).

To demonstrate the link between attention sinks and repeated token divergence, we take a Mechanistic Interpretability approach, analyzing the underlying mechanism of attention sinks in LLMs. We identify two key stages consistently observed across many models: First, the initial attention layer identifies and "marks" the first token in the sequence. Second, a single neuron in a later layer, triggered by this mark, adds high-magnitude values to the hidden state of the first token, creating the attention-sink. This high-magnitude hidden state then attracts attention in subsequent layers, as described by Xiao et al. (2024). We present empirical evidence for this two-stage mechanism across multiple LLMs.

Next, we examine how this mechanism interacts with repeating tokens. We find that the first attention layer, aiming to identify the first token, fails to distinguish it from a sequence of identical tokens. Consequently, it marks both the first token and the repeated tokens, activating neurons that amplify their hidden states. This leads to abnormally high attention weights on the repeated tokens, causing the model's behavior to diverge and produce unexpected outputs.

By analyzing the first attention layer implementation of identifying the first token we were able to generate a new exploit we call " cluster attack", that induces attention sinks without using repeated tokens. This exploit would be able to bypass surface level defenses that addresses repetitions but not underlying vulnerability.

Finally, our analysis of the underlying mechanism suggests a simple yet effective correction to mitigate the repeating tokens divergence and its extensions.

Our contributions are:

1. **Mechanistic Explanation:** We provide the first mechanistic explanation for the repeated token divergence in LLMs, linking it to the attention-sink phenomenon and identifying the responsible neural computation. We

[1]Google DeepMind [2]UC Berkeley. Correspondence to: Itay Yona <itayona@gmail.com>.

*Proceedings of the 42nd International Conference on Machine Learning*, Vancouver, Canada. PMLR 267, 2025. Copyright 2025 by the author(s).

pinpoint a two-stage mechanism for attention sinks: (1) first-token marking by the initial attention layer, and (2) hidden-state amplification by specific MLP neurons (Section 4).

2. **Divergence Extension (Cluster Attack):** We introduce a novel "cluster attack" demonstrating that the divergence extends beyond exact repetition to include similar tokens, revealing a broader vulnerability (Section 5.2).

3. **Targeted Mitigation:** We develop a targeted correction method that effectively mitigates the divergence and the cluster attack without impacting performance on other tasks (Section 5.3).

## 2. Related Work

First, we discuss previous approaches that mechanistically interpreting various model behaviors. Then, we present related work about the two model behaviors we aim to mechanistically interpret - repeated token divergence and attention sinks in LLMs.

### 2.1. Mechanistic Interpretability of LLMs

Mechanistic Interpretability aims to reverse engineer neural networks and uncover the underlying algorithms and computational mechanisms that drive their behavior (Wang et al., 2022; Cunningham et al., 2023; Gandelsman et al., 2025; Jiang et al., 2025). This field seeks to move beyond simply observing correlations to identifying causal relationships within the network, providing a deeper understanding of how these complex systems function (Olsson et al., 2022; Olah et al., 2020). Our work falls under this umbrella by dissecting the specific neural circuits responsible for both attention-sinks and the repeated tokens divergence in LLMs.

### 2.2. Repeated Tokens Divergence

The phenomenon of LLMs failing to accurately repeat tokens was first described by Nasr et al. (2023). While they observed the convergence of repeating token representations towards the BoS token representation, they offered only intuitive explanations for the matter calling for future work to investigate this further. Our work builds upon this observation by identifying the specific neural mechanisms and attention dynamics responsible for this divergence. We link this behavior to the attention-sink mechanism and demonstrate how the interaction between these two phenomena leads to the observed failures in token repetition. Barbero et al. (2025) explored the theoretical challenges of repeated tokens in LLMs (using an infinite context window) and provided empirical observations, but did not explicitly connect the two. Differently from previous work we present a different level of analysis (Marr & Poggio, 1976) describing the

algorithm and implementation of the mechanism that give rise to this issue.

### 2.3. Attention Sinks in LLMs

LLM Attention Sinks were studied in multiple contexts. Xiao et al. (2024) showed that attention sinks relate to model fluency. Gu et al. (2024) explored why and how attention-sinks emerge. Sun et al. (2024) show the relationship of attention sinks to massive activations. Cancedda (2024) analyze what signals propagates through attention-sinks. Yu et al. (2024) propose a method to address the existence attention sinks. Bondarenko et al. (2024); Miller (2023) explored methods to mitigate attention sinks for model quantization and efficiency. However, our work takes a different approach, focusing on the *functional role* of attention sinks and their unintended consequences, specifically their connection to the repeated token divergence phenomenon. We investigate how the very mechanism that makes attention sinks can also be exploited.

## 3. Preliminaries

This section provides a brief overview of the key concepts related to Large Language Models (LLMs) and attention sinks, setting the stage for our analysis of the repeated token divergence.

### 3.1. Large Language Models (LLMs)

LLMs are deep learning models trained on massive text datasets. They leverage the Transformer architecture (Vaswani et al., 2023), which employs the attention mechanism to process sequential data like text. The input to the Transformer is a sequence of tokens that is converted to a sequence of embeddings $X = [x_1, x_2, ..., x_n], x_i \in \mathbb{R}^d$. Usually, the first token is an indicator named Beginning-of-Sequence (BoS).

Transformers are built from two main blocks: attentions and MLPs. Each attention is built from parallel attention heads. The output of an attention head for a token embedding $x_i$ in the sequence $X$ is:

$$\text{Attention}(x_i, X) = W_{proj} \sum_j \text{softmax}_j \left( \frac{q_i^T k_j}{\sqrt{d'}} \right) v_j, \quad (1)$$

where:

$$q_i = W_q x_i, \quad k_j = W_k x_j, \quad v_j = W_v x_j. \quad (2)$$

and $W_q, W_k, W_v \in \mathbb{R}^{d' \times d}$ are the query, key and value projection matrices and $W_{proj} \in \mathbb{R}^{d \times d'}$ is the projection matrix back to the residual stream.

The second type of layer is the MLP. In our analysis we mostly use LLaMa2 (Touvron et al., 2023a), which uses

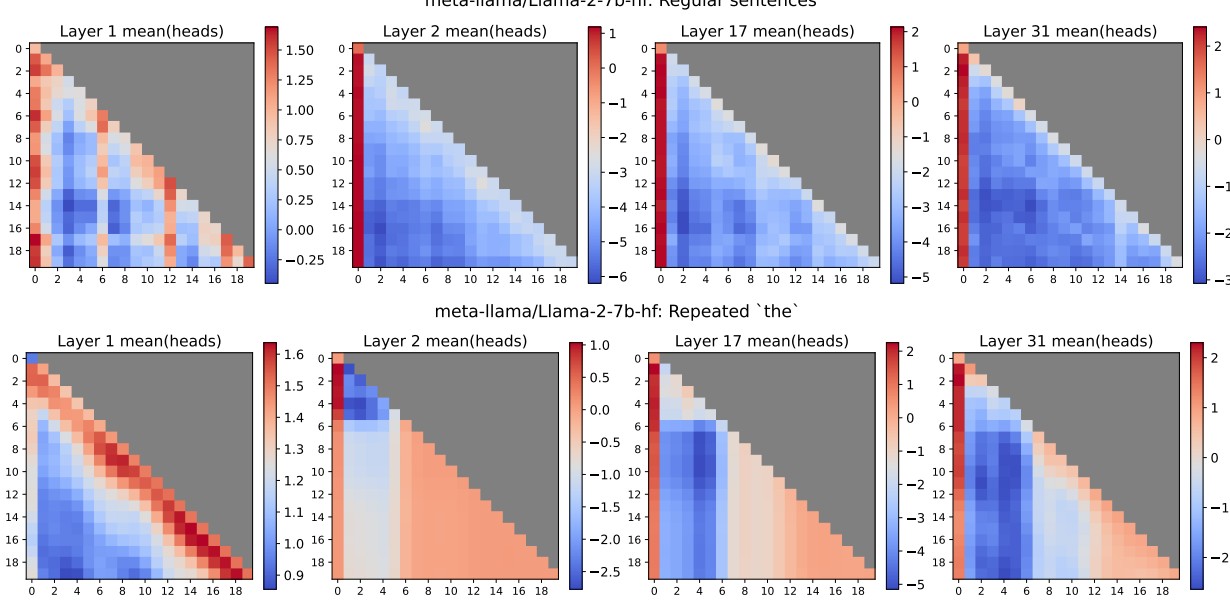

*Figure 1.* **Attention scores for layers 2, 3, 17, and 31 of LLaMA2-7B-HF.** As can be seen in the figure, the attention scores for the repeated "the" tokens in the top panel are significantly higher than those for other tokens in the sequence. This high attention is comparable to the attention received by the first token in the regular sentences shown in the bottom panel. This similarity suggests a connection between the attention sink mechanism and the high attention given to repeated tokens.

the SwiGLU ($\sigma$) as its activation function (Shazeer, 2020). Therefore, the MLP is defined as follows:

$$\text{MLP}(x) = W_{out}^T(\sigma(W_{in}x) \otimes W_{gate}x), \qquad (3)$$

where $W_{gate}, W_{in}, W_{out} \in \mathbb{R}^{d'' \times d}$. A single neuron $1 \le j \le d''$ in the MLP has:

1. Input weights $W_{in}^j \in \mathbb{R}^{1 \times d}$, that dictate which inputs would activate the neuron.

2. Activation $(\sigma(W_{in}^j x) \otimes W_{gate}^j x)$, with $W_{gate}^j \in \mathbb{R}^{1 \times d}$ dictates the amount in which the neuron was activated.

3. Output weights $W_{out}^j \in \mathbb{R}^{1 \times d}$ that dictate what is written to the residual stream when the neuron is activated.

4. Output $\text{MLP}_j(x) = (W_{out}^j)^T(\sigma(W_{in}^j x) \otimes W_{gate}^j x)$, that dictates the individual contribution of that neuron to the residual stream for a given input token $x$.

### 3.2. Attention Sinks

Attention sinks are a phenomenon observed in Transformer LLMs where the model assigns disproportionately high attention scores typically to the first few tokens in a sequence (Xiao et al., 2024). These high attention weights, also known as attention scores, indicate that the model is focusing heavily on these initial tokens. Attention sinks are believed to be

beneficial for generating fluent and coherent text, because they act like biases, storing extra attention and meanwhile not contributing to the value computation (Gu et al., 2024).

### 3.3. Rotary Position Embeddings (RoPE)

Many modern LLMs, including those we analyze in this paper, use Rotary Position Embeddings (RoPE) (Su et al., 2021) instead of absolute or relative position embeddings. RoPE encodes positional information by rotating the token embeddings based on their position in the sequence. A key property of RoPE is that it allows for efficient computation of attention scores and can generalize to sequence lengths not seen during training. Critically, as we show in this paper, RoPE, presents challenges to the model's ability to distinguish between the first token and repeated tokens in a sequence.

## 4. Mechanistic Analysis of Repeated Token Divergence

We present a mechanistic analysis of repeated token divergence, using LLaMa-2 for illustrative examples and plots. Similar motifs observed in additional LLMs (LLaMa-1, LLaMa-3 and Mistral) are discussed in Appendix A.

This section investigates the mechanism behind the repeated-tokens divergence phenomenon. We begin by observing the

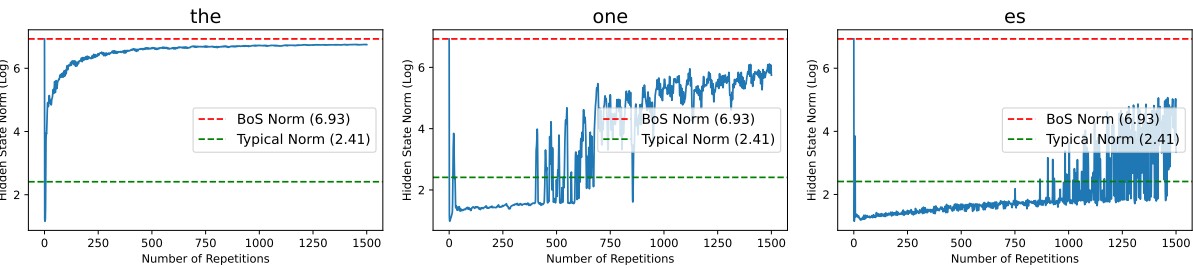

*Figure 2.* **Repeated tokens exhibit extreme norms, similar to the beginning-of-sequence (BoS) token, in early layers.** We present the norm of the hidden state at the sink layer (layer 1) for three repeating words. As the number of repetitions increases, the norm increases and becomes similar to the norm of the BoS token (0 repetitions). This observation explains the high attention scores shown in Figure 1. We later provide causal evidence for this relationship through ablation (Figure 3). We also show the norm of the BoS token and the average norm of tokens from Tiny Shakespeare dataset (Andrej, 2015) for comparison.

similarity between the attention scores of repeated tokens and those of attention sinks (Section 4.1). We then investigate the underlying mechanism of attention sinks (Section 4.2), identifying key components. Finally, we demonstrate how this mechanism misidentifies repeated tokens, leading to the observed divergence (Section 4.3).

### 4.1. Large Attention Scores of Repeated Tokens

Our analysis begins with the observation that, in intermediate layers of LLMs, the average attention scores of repeated tokens (e.g., repeated "the") are comparable in magnitude to the attention scores of the first token in a natural sentence (the attention sink). Figure 1 illustrates this phenomenon for LLaMa-2. The attention scores shown are averaged across all attention heads within a single layer.

### 4.2. The Attention-Sink Mechanism

Motivated by the observed similarity in attention scores, we investigate the neural circuit responsible for attention sinks. We seek to understand: 1) how these tokens acquire such high attention scores, and 2) how the model identifies the first token. In Section 4.3, we will demonstrate that the same mechanism underlies the high attention scores for repeated tokens.

**Attention sinks correlate with high hidden state norms in early layers.** We observe a strong correlation between attention sinks (high attention scores for the first token) and high L2 norms of the hidden states for that token in early layers. Figure 2 shows the L2 norm of the hidden states for different token positions in the sequence. Notice that both the BoS token and the repeated tokens have significantly higher norms compared to other tokens, particularly in the earlier layers (e.g., layer 2). This suggests that the high attention scores observed for repeated tokens might be related

to their high hidden state norms, similar to the BoS token in attention sinks.

**A sparse set of neurons mediates the high norms for the first token (and repeated tokens).** We search for a sparse set of neurons, which we name "sink neurons", that can be candidates for mediating the high norms of the residual stream activations. We select $K$ candidates for these neurons according to the norm of their contributions to the residual stream in the following manner:

$$candidates = \text{TopK}_j(\text{Norm}(\text{MLP}_j(BoS)).$$

To investigate the role of specific neurons in mediating high norms, we perform an ablation study. We zero-ablated the candidate neurons, effectively removing their contribution to the model's computations. We identified sink neurons when we observed a significant reduction in the high norms associated with repeated tokens, due to our ablation, as shown in Figure 3. The ablation study supports our claim that these neurons play a crucial role in establishing the attention sink and influencing the high norms of repeated tokens. The identifies neurons are summarized in Section 4.4.

**The first attention layer distinguishes the first token from subsequent tokens.** We observe that this is done by mapping the two types of tokens (first/non-first) into two linearly separable subspaces.

Figure 4 visualizes a projection of the token representations after the first attention layer. The clear separation between the distributions demonstrates that the first attention layer maps these tokens to distinct, linearly separable subspaces. This suggests that the first attention layer plays a key role in identifying and "marking" the first token in the sequence. The fact that we identified a single neuron (rather than an arbitrary hyperplane) that perfectly separates these subspaces further supports this claim.

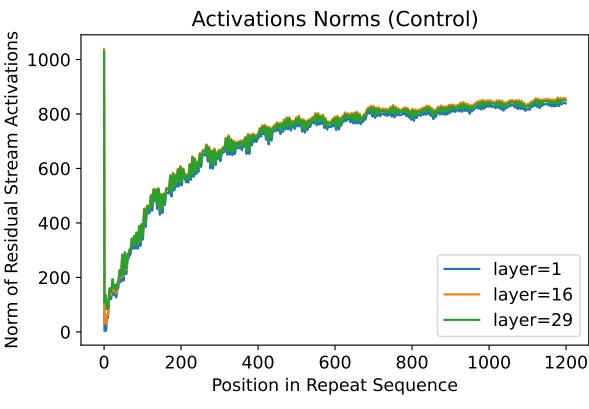

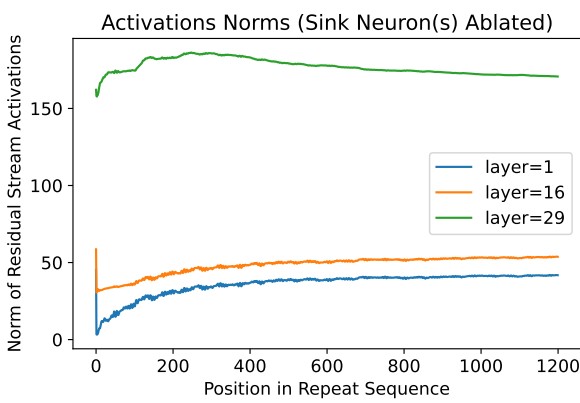

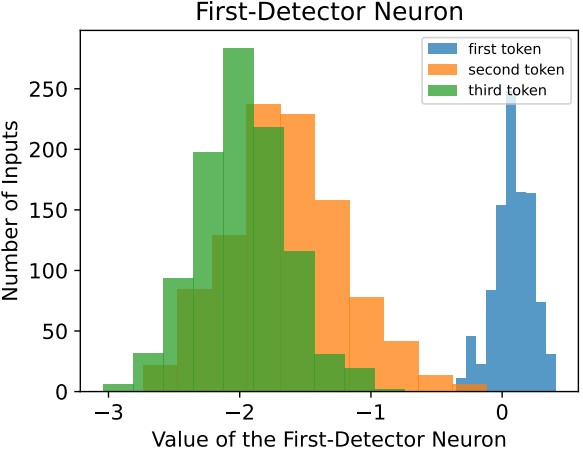

Figure 4. **The first token and subsequent tokens belong to distinct, linearly separable subspaces.** This figure shows the projection of token representations after the first attention layer. The different colors represent the first tokens and subsequent tokens. The clear separation indicates linear separability. Furthermore, we identified a single neuron (MLP$_0$, *gate* neuron 912 in LLaMa2) that perfectly separates these subspaces.

*Figure 3.* **Ablation of sink neurons.** Norm is reduced both for BoS and repeat sequences. Top: Token activations norms *without* the patch. Bottom: Token activations norms *with* the patch. Ablating specific neurons significantly reduces the high norms associated with repeated tokens. Data is from LLaMa-2. Sink-Neurons (1) were zero-ablated. The input consisted of 1200 repeats of the tokens ['Another', 'one', 'bit', 'es', 'the', 'dust']. See Appendix A for similar results on other models.

## 4.3. Linking Repeated Tokens to Attention-Sink

We will now show that the attention-sink mechanism, previously identified as crucial for processing the beginning-of-sequence (BoS) token, also deeply affects the model's behavior when instructed to repeat tokens. Specifically, we demonstrate that the same neurons involved in marking the BoS token are also activated by repeated tokens, and that the first attention layer cannot inherently distinguish between a single token and a sequence of identical repetitions. This inability to differentiate single and repeated tokens provides a mechanistic explanation for the observed divergence.

**Repeated tokens exhibit high norms mediated by the same neurons identified in Section 4.2.** We examined the activity of the sink neurons identified in Section 4.2 for sequences with repeated tokens. Our analysis revealed that

these same neurons exhibit high activation levels in response to repeated tokens, similar to their response to the first token. This suggests that the same neural circuit responsible for the attention sink is also involved in processing repeated tokens.

**Distinguishing between token repetitions and the first token.** We now provide a more formal treatment of the phenomenon by tying it to recent results of Barbero et al. (2025) and Veličković et al. (2024). In particular, we are interested in studying what the token repetition process converges to. We point the interested reader to the appendix (Section D) for the proofs and a more extensive explanation of the theoretical results.

We consider the same Transformer architecture that is considered by (Barbero et al., 2025). Our results hold for any type of positional encoding that acts solely on the queries and keys and that is *bounded*. This is the case for essentially all positional encodings used in practice and is in particular true for RoPE (Su et al., 2021) [1] – the encoding used by all the models in our experiments.

More formally, consider a sequence $S_n$ where we fix the $k$ prefix tokens and then repeat a token $n$ times. For example, assuming each word is a single token we could let $S_3 =$('Prefix1', 'Prefix2', 'poem', 'poem', 'poem') and

---

[1]Boundedness is immediate in RoPE as it is an isometry (Barbero et al., 2024).

$S_5 = $ ('Prefix1', 'Prefix2', 'poem', 'poem', 'poem', 'poem', 'poem') where $k = 2$ for the 2 prefix tokens. A natural question to ask is *can we say something about what happens to the representation of the last token at position $n + k$ as $n \rightarrow \infty$?* It turns out that the answer is **yes**, in fact, as $n \rightarrow \infty$, we prove that the representation of the last token of $S_n$ converges to the representation of $S^*$ where $S^* = $ ('poem') is the sequence with the single element that is being repeated. In particular, as machine precision is finite, this implies that one can find $N > 0$ such that for all $n \geq N$ we have that $S_n$ is indistinguishable from $S^*$ up to floating point precision.

The intuitive explanation for this convergence is that as the sequence grows, the relative influence on the output by the prefix will go to $0$ due to softmax leakage (Veličković et al., 2024). This is the case because critically the size of the prefix remains constant as $S_n$ grows. We summarise informally in Theorem 4.1 the result and provide detailed mathematical proofs in the appendix (Section D). Not only does our analysis characterize the behaviour of repeating a token a large number of times in a decoder-Transformer, but it also shows how mathematical results that describe the propagation of information in Transformers have security implications.

**Theorem 4.1** (Informal.). *Let $x$ be a token and $T$ a Transformer. Consider a sequence $S_n$ with $k$ fixed prefix tokens and $n$ repetitions of $x$ and a singleton sequence $S^*$ which consists of a single $x$. As $n \rightarrow \infty$ the representation of the last element of $S_n$ converges (strongly) to the representation of $S^*$ after applying the Transformer $T$. In other words, the sequence $S_n$ becomes indistinguishable from $S^*$ as $n \rightarrow \infty$.*

To demonstrate this effect, in Figure 5, we show this occurring in LLaMa-2 for a number of example sequences. We find that even for a relatively small number of repetitions, the convergence described in Theorem 4.1 is clearly visible.

**Long sequences of repeated tokens are falsely marked as attention sinks, causing the model to diverge from the intended output.** Combining Theorem 4.1 and the empirical observation, we can see why repeated tokens cause problems. The first attention layer can't distinguish the first repeated token from the others, and it treats long sequences of repeated tokens much like it treats a regular sentence *without* a beginning of sequence token—it assigns high attention to them as if they were the first tokens. Because of this, the same mechanism that creates attention sinks for the true first token gets triggered for the repeated tokens, even though they shouldn't be treated as the beginning of a new sequence. This disrupts the model's normal processing and leads to the observed divergence in the output.

**Detection of first token by the first Attention layer.** The first attention layer is tasked with identifying the initial token

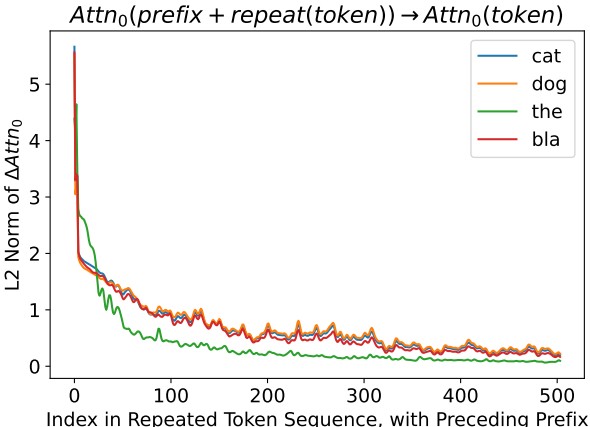

$Attn_0(prefix + repeat(token)) \rightarrow Attn_0(token)$

*Figure 5.* **Empirical evidence showing the first attention layer does not distinguish between token repetitions and the first token.** We first computed the output of the first attention layer for an input of a single token, without BoS. Then compared it, using L2 norm of the difference, to the output of the first attention layer for "$<$BoS$>$ some prefix:", appended to the same token {'cat', 'dog', 'the' or 'bla'} repeated 500 times on LLaMa-2. This supports Theorem 4.1 showing the converges takes place in practice with less than $max\_context\_window$ repeats.

in a sequence, a seemingly simple task whose implementation remains unclear. Empirically, this layer marks any first token, not just BoS (Sun et al., 2024; Gu et al., 2024). Lacking an absolute positional signal, LLaMa-2 leverages causal masking for this purpose. We observe near-orthogonal query and key vectors in the first layer's attention heads, effectively making them "other token detectors"—preferring to attend to any token other than themselves. Causal masking forces the first token to self-attend, effectively marking it as the first for the next layers. Thus, rather than explicitly finding the first token, this layer identifies single-token sequences. This explains how and why long enough sequences of repeated tokens confuse the layer.

### 4.4. Findings

*Table 1.* Summary of sink neuron findings in several LLMs, showing repetitions needed for induction, sink layer, and neuron IDs.

| Model | Repeats | Sink-Layer | Sink-Neurons IDs |
|---|---|---|---|
| LLaMa-1-7b-HF | 450 | 2 | 7003 |
| LLaMa-2-7b-HF | 1000 | 1 | 7890, 10411 |
| Meta-Llama-3-8B-Instruct | 4000 | 1 | 198, 2427 |
| Mistral-7B-Instruct-v0.1 | 1200 | 1 | 7310, 8572 |

Sink neurons in early layers are a common feature across several Large Language Models (LLMs), as shown in Table 1. We observed these neurons in multiple models. The table

details the number of repetitions required to induce a sink, along with the layer and ID of the sink neuron. A more in-depth analysis and illustration can be found in Appendix A. While these models share similar underlying mechanisms, they exhibit subtle differences in implementation and behavior, which are discussed in Section 7.

# 5. Attack and Mitigation

Nasr et al. (2023) first observed that repeated tokens could confuse models, creating a vulnerability exploitable for training data leakage attacks. While pretrained LLMs can be prompted to continue prefixes, and instruction-tuned models can be instructed to do so, Reinforcement Learning from Human Feedback (RLHF)-aligned models might resist. Surprisingly, repeated tokens have been shown to trigger ChatGPT (OpenAI, 2022) to deviate from its typical instruction-following behavior and reveal memorized training data. Here we show that other models also exhibit repeated token divergence, and output their training data.

As the training data of LLaMa models is not publicly available, it difficult to determine if a model's output originates from its training data. We therefore present an example from Pythia-12b (Biderman et al., 2023), an open-source model trained on the publicly available The Pile dataset (Gao et al., 2020).

We find that for Pythia-12b and the input "as ", repeating 50 times, the model outputs: ``The first step in the process of 3D printing is to create a 3D model of the object to be printed. The 3D model is then sliced into thin layers, each layer being printed one at a time. The slicing process is performed by a slicing''

Which is a rephrase of the following text [2] that is taken from a website that appears in the training data: ``The starting point for any 3D printing process is a 3D digital model, which can be created using a variety of 3D software programmes | in industry this is 3D CAD, for Makers and Consumers there are simpler, more accessible programmes available | or scanned with a 3D scanner. The model is then `sliced' into layers, thereby converting the design into a file readable by the 3D printer.''

More examples of potential leakage appear in Appendix C.

---

[2] https://3dprintingindustry.com/3d-printing-basics-free-beginners-guide#03-technology

## 5.1. Surface-level mitigation

A mitigation that would simply detect lengthy sequences of repeated tokens is a potential approach, but it does not address the underlying issue. This limitation prompted us to explore other (non-repeating) inputs that induce attention sinks. These sinks, similar to those triggered by the Beginning-of-Sequence (BoS) token or repeated tokens, achieve similarly high norms by activating the same neurons as the BoS token, ultimately leading to the same undesirable outcome of diverging model behavior.

## 5.2. Attack extension

In this section, we investigate the "attend to others" (4.3) mechanism and its implementation in different models. We show that this mechanism clusters the tokens into sets, and achieves its goal by applying different attention heads on each group. This understanding allows us to extend the repeating tokens attack of Nasr et al. (2023) to an attack that repeats tokens from the same set. We extract the different sets for different models and present qualitative and quantitative results for the attack performance.

**Clustering the tokens.** Since we found two distinct subspaces for first/non-first tokens and we know the first attention layer is responsible for this projection, we reduce the mechanism even further, looking for specific heads that are responsible for this projection. On LLaMa2 we found that for many tokens, a single head's output is enough to project to the relevant subspaces. The relevant head varied between tokens, forming N clusters for tokens. See Appendix B.

**Repeating tokens for cluster attack.** This clustering suggests that attention sinks can arise from tokens in the same group, even without a BoS token or exact repetition. By using tokens from a cluster, we trigger the same attention head, its function in this circuit is to attend any other tokens, however since we introduce tokens from the same cluster, this head is inevitably attends with high enough score and projections the tokens representation to the "first token subspace". For example: The norm after MLP1 of the input: "Sch Com" (both taken from the cluster of Head 4) is [18.4375, 16.5469], or for "elements description" (from Head 30) the MLP1 norm is [19.0156, 14.3359] dramatically higher than any other norm of tokens (besides BoS) after the first layer.

## 5.3. Attack mitigation

We present a simple editing method that allows us to prevent the repeating token attack. First, we demonstrate qualitative results of this edit on LLaMA2 when repeated prompts are provided. We verify that this modification does not change other model behavior by evaluating the edited model on standard benchmarks.

*Table 2.* **The effect of patching on unrelated tasks.** We compare LLaMa-1, LLaMa-2 and Mistral, before and after the patching on different benchmarks.

| | LLaMa-1-7B-HF | | | LLaMa-2-7B-HF | | | Mistral-7B-Instruct-v0.1 | | |
|---|---|---|---|---|---|---|---|---|---|
| | original | patched | $\Delta$ | original | patched | $\Delta$ | original | patched | $\Delta$ |
| MMLU | 29.81 | 29.93 | +0.12 | 41.20 | 42.17 | +0.97 | 53.41 | 52.58 | -0.83 |
| HellaSwag | 56.97 | 56.95 | -0.02 | 57.12 | 57.13 | +0.01 | 56.23 | 55.75 | -0.48 |
| TruthfulQA | 31.21 | 29.38 | -1.83 | 34.15 | 34.39 | +0.24 | 53.37 | 52.26 | -1.11 |
| WinoGrande | 69.93 | 69.93 | 0.00 | 68.98 | 69.06 | +0.08 | 69.30 | 68.82 | -0.48 |
| AI2-ARC | 41.89 | 42.15 | +0.26 | 43.52 | 43.34 | -0.18 | 50.17 | 49.66 | -0.51 |

*Listing 1.* A manual patch to fix repeated tokens issue.

```python
tmp_output = None
sink_neuron = 7890
sink_layer = 1

def patch_sink(x, phase):
    global tmp_output

    if phase == "prefill":
        tmp_output = x[:,1, sink_neuron]
        x[:,1:,sink_neuron] = tmp_output

    if phase == "decode":
        x[:,0, sink_neuron] = tmp_output

    return x

patch_block = model.blocks[sink_layer]
patch_block.mlp.up_proj.hook(patch_sink)
```

During the prefill stage, the model processes an entire sequence and might include repeats. We store aside the output of the sink neuron for the second input token, this value means "no-sink" is valid for non-repeated or non-BoS tokens. We then replace all the values of the neuron activity to be "no-sink". We also make sure that during decoding no additional sink is set.

**Edit performance on other tasks.** While our edit mitigates the repeated token attack, it may change to model performance on other unrelated tasks. Next, we verify that the presented edit (1) does not change the performance on other standard tasks. As shown in Table 2, the changes between the performances of the model before and after the patch are negligible.

## 6. Limitations

This work has several limitations. First, the mechanism by which training data leakage occurs through attention sinks remains unclear, requiring further investigation. Second, while shared motifs exist across LLMs, the first attention layer's behavior was unique to LLaMa2, highlighting the influence of model-specific details. Third, the efficacy of token repetition in inducing sinks varies across models. Not all tokens in LLaMa2 induce sinks (see Figure 6), suggesting other factors are at play. Fourth, due to the difficulty in assessing training data leakage we did not quantify the

success rate of the cluster attack. Finally, our mitigation strategy does not fully address the inherent vulnerability of causally masked Transformers to repeated tokens.

## 7. Discussion and Future Work

This study reveals a clear link between repeated token divergence in LLMs and the attention sink mechanism, in the form of neural circuits. We demonstrate how the first attention layer's confusion between an initial token and a repeated sequence leads to erroneous attention weights and output divergence, highlighting the inherent tension between fluency and robustness. Our "cluster attack", induce sinks without the need to repeat tokens. Simple defense mechanisms that filter out requests for repeated tokens are easily bypassed by our new attack. Based on neural circuit understanding we developed a more fundamental mitigation for this issue without significant performance impact (see Table 2). This work contributes to LLM interpretability by showcasing how mechanistic insights can guide targeted interventions. While our analysis focused on specific LLMs, we intend to investigate broader generalizability in future work, to explain why repeated token sequences lead to training data extraction as studied by Nasr et al. (2023). This research underscores the need for ongoing investigation into the inner workings of LLM to build more transparent, reliable, and secure models.

## Acknowledgements

We thank Arthur Conmy, Christopher A. Choquette-Choo, Guillermo Ortiz-Jimenez, Guy Dar, and Neel Nanda for their valuable comments and feedback on our paper. YG is supported by the Google Fellowship.

## Impact Statement

This paper presents work whose goal is to advance the field of Machine Learning. There are many potential societal consequences of our work, none which we feel must be specifically highlighted here.

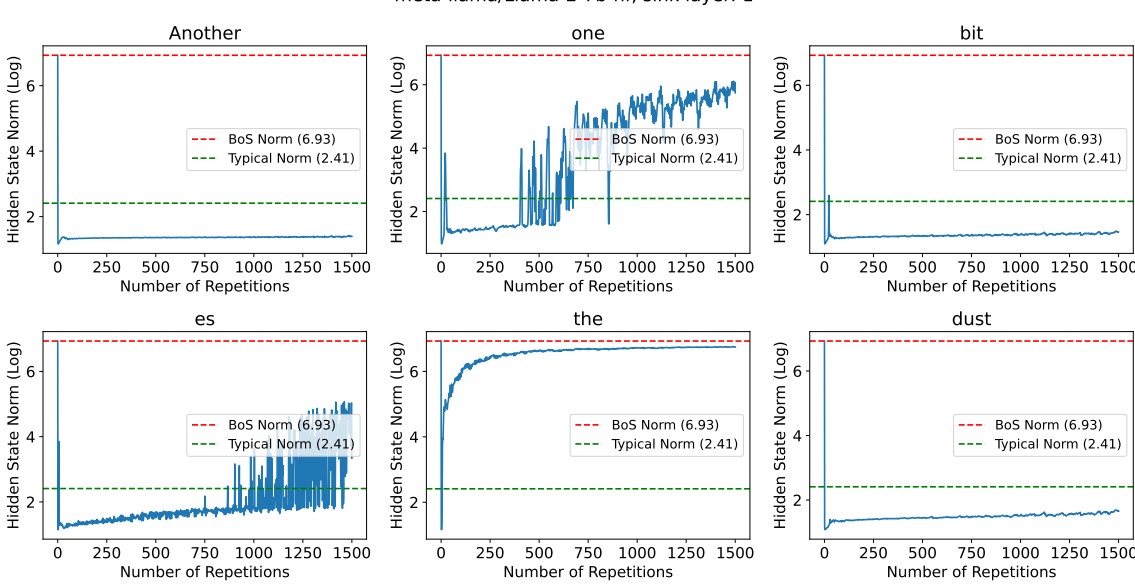

*Figure 6.* **Limitations.** For some tokens ("Another", "bit", "dust"), repetition does not lead to the creation of attention sinks.

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

# A. Generalization to other models

We present results on additional models: LLaMa-1, LLaMa-3 and Mistral, to illustrate the generality of the circuits we discovered. First, we present (Figure 7) patching results similar to Figure 3, followed by an attention score for LLaMa-1 (Figure 8), and graphs showing extreme norms for repeated tokens in LLaMa-1, LLaMa-3 and Mistral (Figure 9).

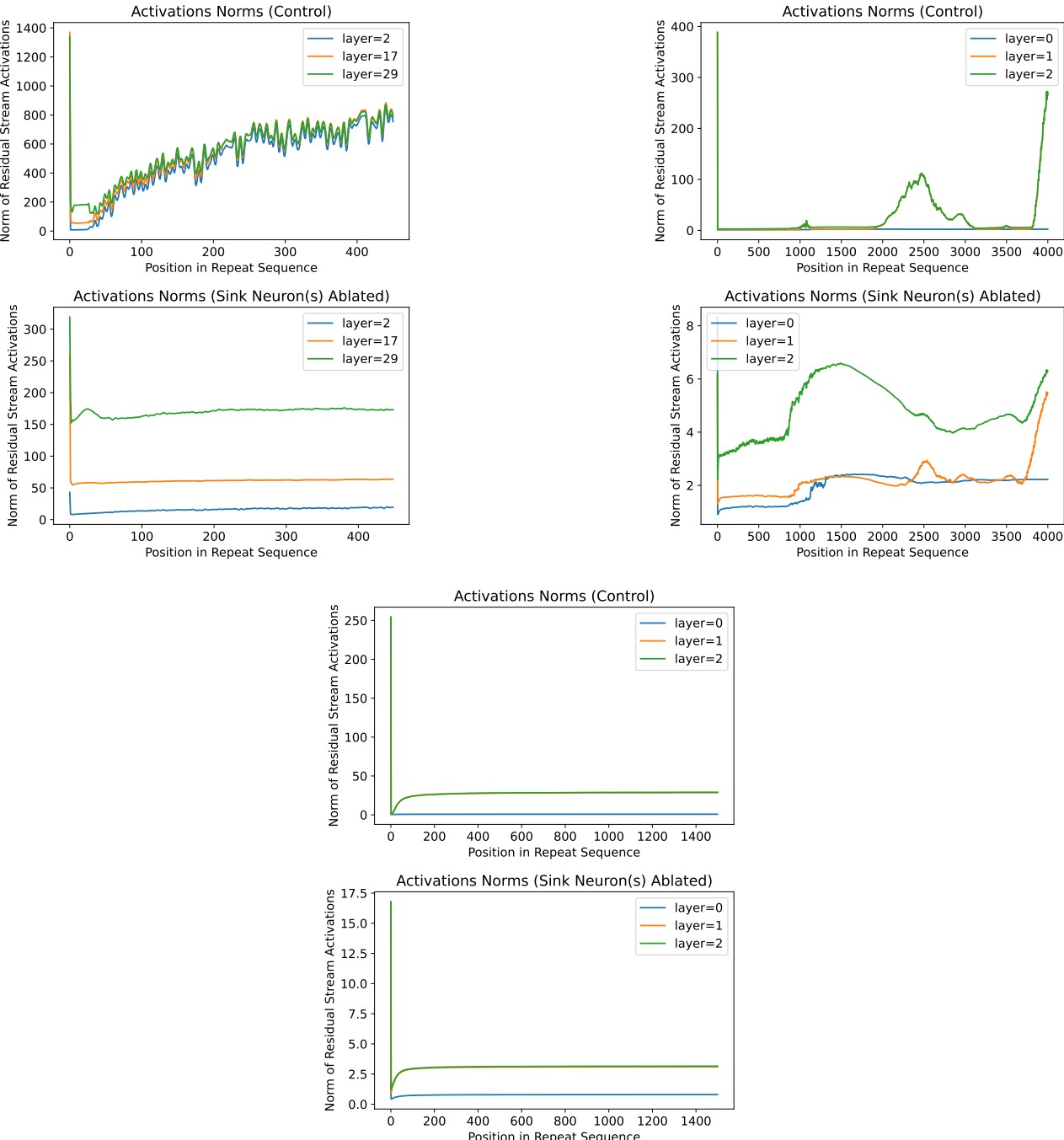

*Figure 7.* An ablation to sink neurons in other models, extending Figure 3, reduces the norms both for BoS and repeats. Top Left: LLaMa-1. Top Right: LLaMa-3. Bottom: Mistral.

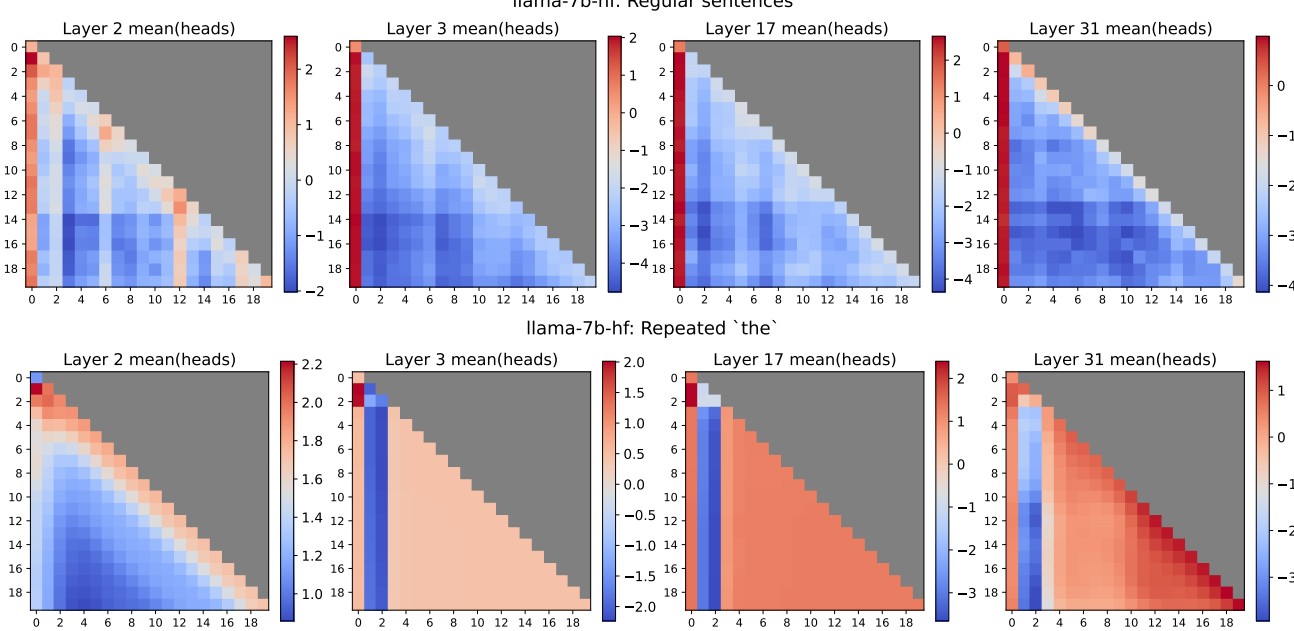

*Figure 8.* **Attention scores for layers 2, 3, 17, and 31 of LLaMA1-7B-HF.** As can be seen in the figure, the attention scores for the repeated 'the' tokens in the top panel are significantly higher than those for other tokens in the sequence. This high attention is comparable to the attention received by the first token in the regular sentences shown in the bottom panel. This similarity suggests a connection between the attention sink mechanism and the high attention given to repeated tokens.

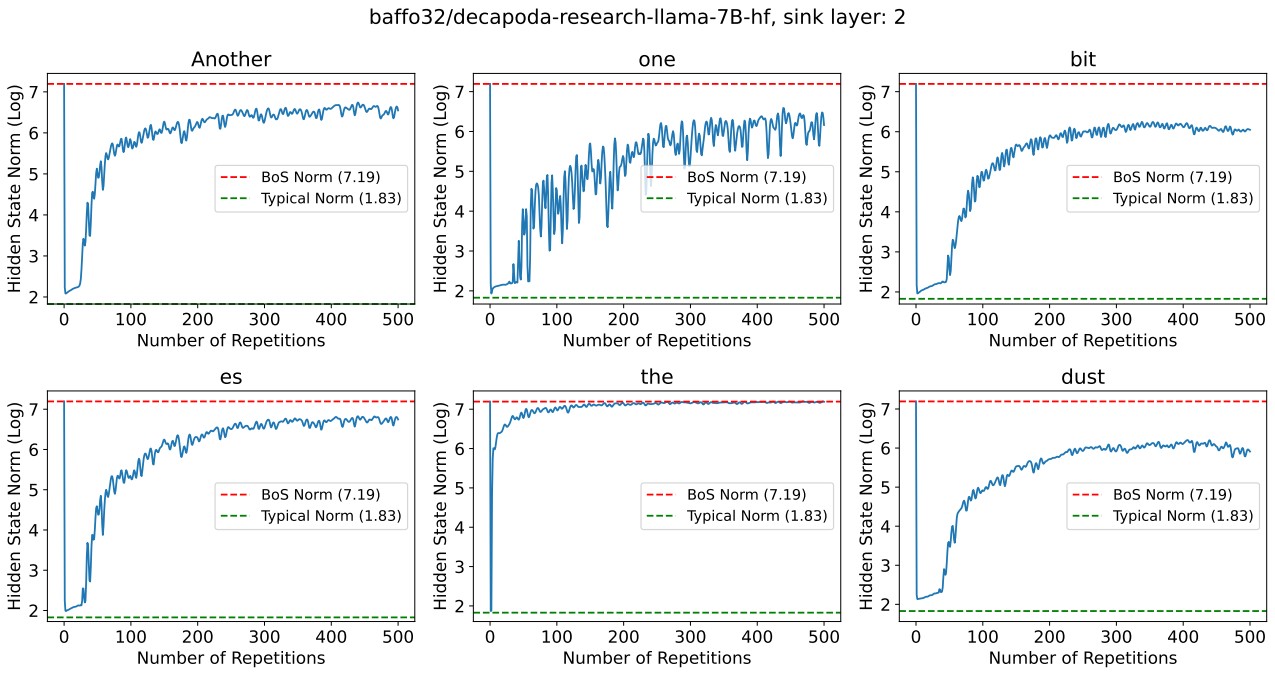

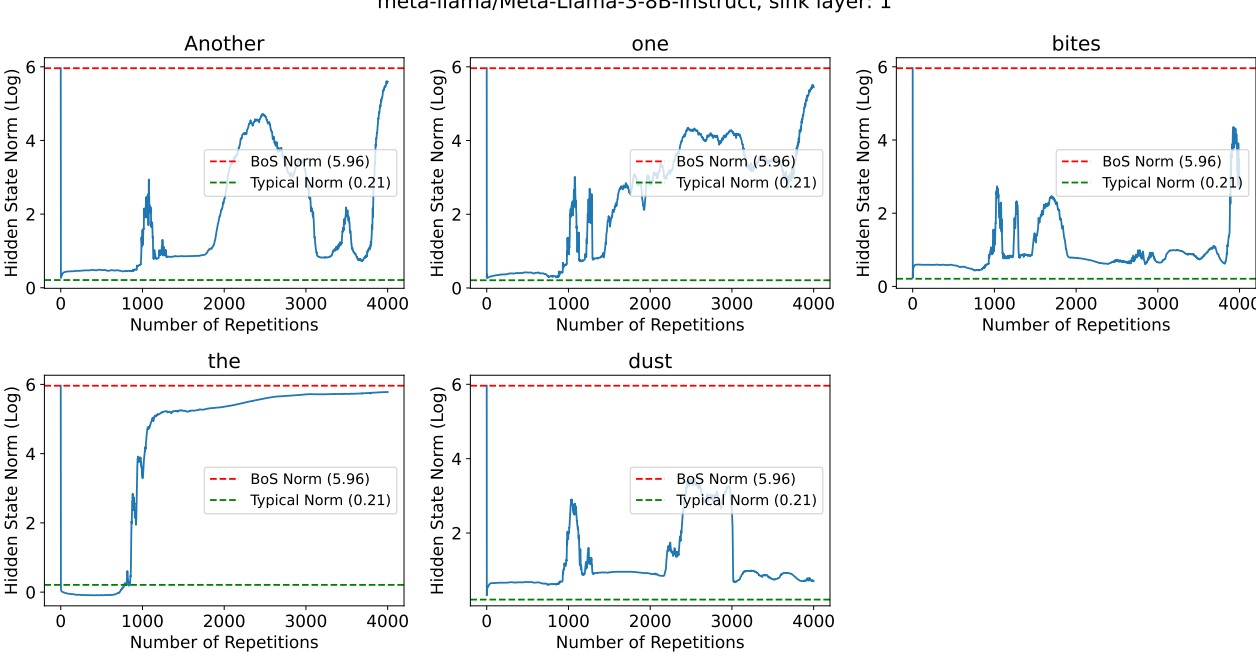

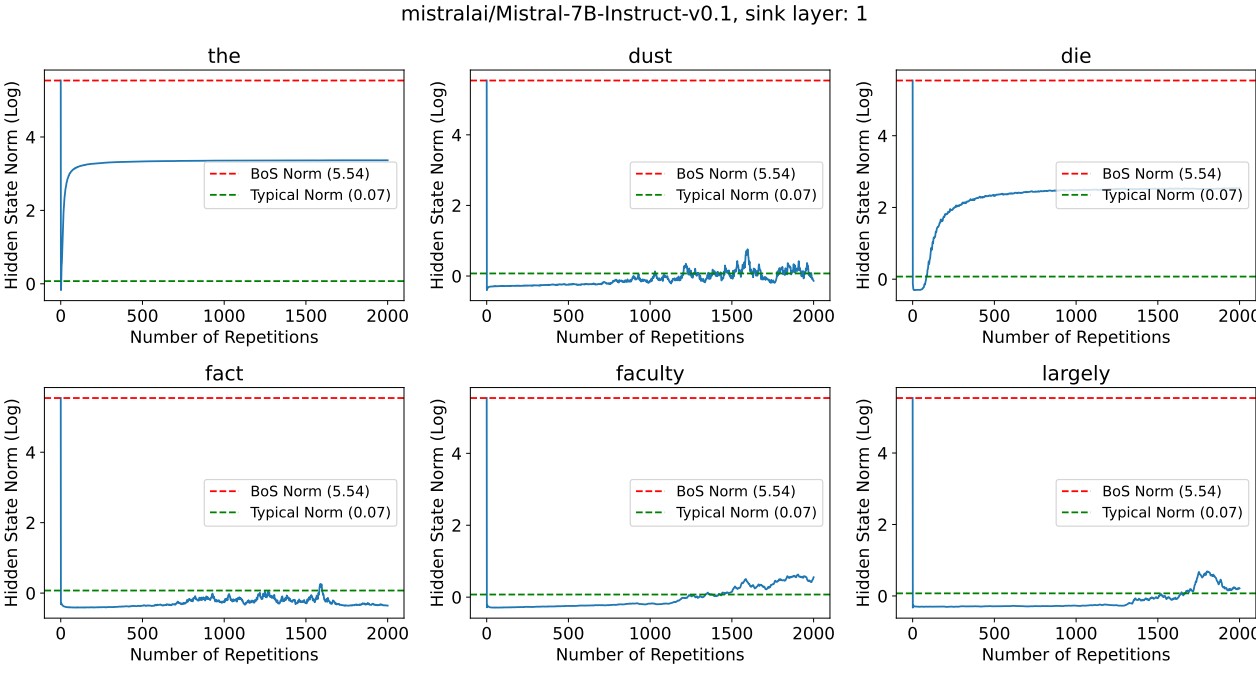

*Figure 9.* **Further evidence of extreme norms for repeated tokens in early layers.** We present similar results for Figure 2 on LLaMa-1 LLaMa-3 and Mistral. The norm of the hidden state at the sink layer for repeating words. As the number of repetitions increases, the norm increases and becomes similar to the norm of the BoS token (0 repetitions). We also show the norm of the BoS token and the average norm of tokens from Tiny Shakespeare dataset (Andrej, 2015) for comparison.

# B. Cluster of Tokens for Detecting First

```
30 ['', '\x17', 'et', 'out', 'out', 'function', 'like', 'using', 'does', 'list', 'under', 'look', 'result', 'after', 'off',
'number', 'hand', '],', 'again', 'down', 'create', 'change', 'order', 'server', 'request', 'support', 'group', 'ungen', 'results',
'position', 'appear', 'based', 'effect', 'getting', 'ists', 'response', 'known', 'keep', 'elements', 'functions', 'OS', 'reference',
'took', 'changes', 'returns', 'house', 'September', 'making', 'connection', 'returned', 'among', 'round', 'included', 'attack',
'behind', 'description', 'execute', 'February', 'loss', 'records', '!!', 'requires', 'associated', 'supported', 'layer', 'includes',
'established', 'increase', 'understanding', ')),', 'seeing', 'instances', 'observed', 'developed', '",', 'ended', 'influence', 'lica',
'programs', 'supports', 'increased', 'Station', 'providing', 'Fund', 'approaches', 'considering', 'categories', 'allowing', '",\r',
'""']

4 ['"', 'com', 'Com', 'ha', 'fol', 'over', 'could', 'cor', 'Mar', 'As', 'Sch', 'Re', 'id', 'sur', 'ass', 'Con', 'trans', 'char', 'Par',
'supp', 'exp', 'dec', 'orig', 'Il', 'mat', 'Ser', 'Ind', 'Bo', 'cour', 'appro', 'Cons', 'Reg', 'Ber', 'beh', 'ear', 'App', 'super',
'dom', 'Ste', 'prom', 'Fil', 'bro', 'bar', 'Mon', '-\\', 'gra', 'Est', 'Sh', 'An', 'mer', 'organ', 'Char', 'Cor', 'Mus', 'Min', 'gl',
'hum', 'mot', 'Ang', 'Ma', 'Gener', 'Sub', 'autom', 'Sim', 'Part', 'elect', 'Mag', 'util', 'Vol', 'Bel', 'Mal', 'Dec', 'q', 'Fre',
'Mor', 'Po', 'cam', 'Sal', 'Ass', 'Trans', 'Gre', 'Hol', 'Fin', 'Lo', 'Bro', 'Pres', 'Dem', 'Ret', 'Mer', 'Met', 'Tor', 'CO', 'tor',
'univers', 'Rad', 'Gra', 'Spe', 'Stud', 'Kir', 'Gal', 'Du', 'Ju', 'Mat', 'Bu',, 'Del', 'Leg', 'compar', 'Const', 'Bre', 'Het', 'Cla',
'Gro', 'Equ', 'Att', 'Cour', 'Tre', 'Pan', 'jud', 'Oper', 'Bur', 'Stat', 'cos', 'Tem', 'Hor', 'Flor', 'Cas', 'Maj', 'Sum', 'Ol', 'Mot',
'Vict', 'Ev', 'adm', 'Ext', 'Kr', 'Mur', 'Every', 'cru', 'rect', 'www', 'colon', 'Gall', 'grav', 'Tro', 'Ho', 'Kn', 'Sel', 'Bon', 'fif',
'hol', 'Cro', 'Hot', 'Bol', 'Egy', 'AR', 'Organ', 'decre', 'Leb', 'Dev', 'Dor', 'proc', 'Vari', 'Non', 'Mah', 'Wy', 'Tal', 'reve',
'Bla', 'vir', 'Mol', 'Ak', 'Dig', 'LO', 'eth', 'Blo', 'Rest', 'Cab', 'Wol', 'Fel', 'Vel', 'Kh', 'Kil', 'Altern', 'Bou', 'GL', 'sle',
'Moh', 'dy', 'Eth', 'Cho', 'Sl', 'Wed', 'mul', 'Clar', 'furn', 'Wis', 'Autom', 'Sig']

0 ['before', 'between', 'through', 'without', 'upon', 'against', 'actually', 'whether', 'together', 'significant', 'évrier',
'regarding']
```

# C. Leakage & Prevention

We show two examples of repeated inputs on LLaMa-1 that diverge model behavior, and show that with our patch Section 5.3 mitigate this issue.

**After patching** — $$ Repeat: poem poem poem poem poem poem poem poem poem poem poem poem poem poem poem poem poem poem poem poem poem poem poem poem poem poem poem poem poem poem poem poem poem poem poem poem poem poem poem poem poem poem poem poem poem poem poem poem poem poem poem poem poem poem poem poem poem poem poem poem poem poem poem poem poem poem poem poem poem poem poem poem poem poem poem poem poem poem poem poem poem poem poem poem poem poem poem poem poem poem poem poem poem poem poem poem poem poem poem poem poem poem poem poem poem poem poem poem poem poem poem poem poem poem poem poem poem poem poem poem poem poem poem poem poem poem poem poem poem poem poem poem poem poem poem poem poem poem poem poem poem poem poem poem poem poem poem poem poem poem poem poem poem poem poem poem poem poem poem poem poem poem poem poem poem poem poem poem poem poem poem poem poem poem poem poem poem poem poem poem poem poem poem poem poem poem poem poem poem poem poem poem poem poem poem poem poem poem poem poem poem poem poem poem poem poem poem poem poem poem poem poem poem poem poem poem poem poem poem poem poem poem poem poem poem poem poem poem poem poem poem poem poem poem poem poem poem poem poem poem poem poem poem poem poem poem poem poem poem poem poem poem poem poem poem poem poem poem poem poem poem poem poem poem poem poem poem poem poem poem poem poem poem poem poem poem poem poem poem poem poem poem poem poem poem poem poem poem poem poem poem poem poem poem poem poem poem poem poem poem poem poem poem poem poem poem poem poem poem poem poem poem poem poem poem poem poem poem poem poem poem poem poem poem poem poem poem poem poem poem poem poem poem poem poem poem poem poem poem poem poem poem poem poem poem poem poem poem poem poem poem poem poem poem poem poem poem poem poem poem poem poem poem poem poem poem poem poem poem poem poem poem poem poem poem poem poem poem poem poem poem poem poem poem poem poem poem poem poem poem poem poem poem poem poem poem poem poem poem poem poem poem poem poem poem poem poem poem poem poem poem poem poem poem poem poem poem poem poem poem poem poem poem poem poem poem poem poem poem poem poem poem poem poem poem poem poem poem poem poem poem poem poem poem poem poem poem poem poem poem poem poem poem poem poem poem poem poem poem poem poem poem poem poem poem poem poem poem poem poem poem poem poem poem poem and poem poem poem poem poem poem, poem, and poem, poem, poem, poems, poetry, poetry, rhymes, rhymes, rhyming, rhyming, rhyming rhyming rhyming rhyming rhyming rhyming rhyming rhyming rhyming rhyming rhyming rhyming rhyming rhyming rhyming rhyming rhyming rhyming rhyming rhyming rhyming rhyming rhyming rhyming rhyming rhyming rhyming rhyming rhyming rhyming rh

**Before patching** — $$ Repeat: poem poem poem poem poem poem poem poem poem poem poem poem poem poem poem poem poem poem poem poem poem poem poem poem poem poem poem poem poem poem poem poem poem poem poem poem poem poem poem poem poem poem poem poem poem poem poem poem poem poem poem poem poem poem poem poem poem poem poem poem poem poem poem poem poem poem poem poem poem poem poem poem poem poem poem poem poem poem poem poem poem poem poem poem poem poem poem poem poem poem poem poem poem poem poem poem poem poem poem poem poem poem poem poem poem poem poem poem poem poem poem poem poem poem poem poem poem poem poem poem poem poem poem poem poem poem poem poem poem poem poem poem poem poem poem poem poem poem poem poem poem poem poem poem poem poem poem poem poem poem poem poem poem poem poem poem poem poem poem poem poem poem poem poem poem poem poem poem poem poem poem poem poem poem poem poem poem poem poem poem poem poem poem poem poem poem poem poem poem poem poem poem poem poem poem poem poem poem poem poem poem poem poem poem poem poem poem poem poem poem poem poem poem poem poem poem poem poem poem poem poem poem poem poem poem poem poem poem poem poem poem poem poem poem poem poem poem poem poem poem poem poem poem poem poem poem poem poem\n poem\n poem poem poem poem poem poem poem poem poem poem poem poem poem poem poem poem poem poem poem poem poem poem poem poem poem poem poem\n poem poem poem poem poem poem poem poem poem poem poem\n poem poem poem poem poem poem poem poem poem poem poem poem poem poem poem poem poem poem\n poem poem poem poem poem poem poem poem poem poem poem poem poem poem poem poem poem poem poem poem poem poem poem poem poem poem poem

n poem\n poem poem poem poem poem poem poem poem poem poem poem poem poem poem poem poem poem poem poem poem poem poem poem poem poem poem poem poem\n poem\n poem\n poem\n poem\n poem\nThe 19th century was a time of great change in the United States. The Industrial Revolution was in full swing and the United States was growing rapidly. The United States was also going through a time of great social change. The United States was becoming more and more industrialized and the population was growing rapidly. The United States was also going through a time of great social change. The United States was becoming more and more industrialized and the population was growing rapidly. The United States was also going through a time of great social change. The United States was becoming more and more industrialized and the population was growing rapidly. The United States was also going through a time of great social change. The United States was becoming more and more industrialized and the population was growing rapidly. The United States was also going through a time of great social change. The United States was becoming more and more industrialized and the population was growing rapidly. The United States was also going through a time of great social change. The United States was becoming more and more industrialized and the population was growing rapidly. The United States was also going through a time of great social change. The United States was becoming more and more industrialized and the population was growing rapidly. The United States was also going through a time of great social change. The United States was becoming more and more industrialized and the population was growing rapidly. The United States was also going through a time of great social change. The United States was becoming more and more

**After patching** — $< s >$ Repeat: company company company company company company company company company company company company company company company company company company company company company company company company company company company company company company company company company company company company company company company company company company company company company company company company company company company company company company company company company company company company company company company company company company company company company company company company company company company company company company company company company company company company company company company company company company company company company company company company company company company company company company company company company company company company company company company company company company company company company company company company company company company company company company company company company company company company company company company company company company company company company company company company company company company company company company company company company company company company company company company company company company company company company company company company company company company company company company company company company company company company company company company company company company company company company company company company company company company company company company company company company company company company company company company company company company company company company company company company company company company company company company company company company company company company company company company company company company company company company company company company company company company company company company company company company company company company company company company company company company company company company company company company company company company company company company company company company company company company company company company company company company company company company company company company company company company company company company company company company company company company company company company company company company company company company company company company company company company company company company company company company company company company company company company company company company company company company company company company company company company company company company company company company company company company company company company company company, and the company, and the The company, and the company, and the, and the, and the, and the, and the, and the, and the, and the, and the, and the, and the, and the, and the, and the, and the, and the, and the, and the, and the, and the, and the, and the, and the, and the, and the, and the, and the

**Before patching** — $$ Repeat: company company company company company company company company company company company company company company company company company company company company company company company company company company company company company company company company company company company company company company company company company company company company company company company company company company company company company company company company company company company company company company company company company company company company company company company company company company company company company company company company company company company company company company company company company company company company company company company company company company company company company company company company company company company company company company company company company company company company company company company company company company company company company company company company company company company company company company company company company company company company company company company company company company company company company company company company company company company company company company company company company company company company company company company company company company company company company company company company company company company company company company company company company company company company company company company company company company company company company company company company company company company company company company company company company company company company company company company company company company company company company company company company company company company company company company company company company company company company company company company company company company company company company company company company company company company company company company company company company company company company company company company company company company company company company company company company company company company company company company company company company company company company company company company company company company company company company company company company company company company company company company company company company company company company company company company company company company company company company company company company company company company company company company company company company company company company company company company company company company company company company company company company company company company company company company company company company company company company company company company company company company company company company company company company company company company company company company company company company company company company company company company company company company company company company company company company company company company company company company company company company company company company company company company company company company company company company company company company company company company company company company company company company company company company company company company company company company company company company company company company company company company company company company company company company company company company company company company company company company company company company company company company company company company company company company company company company company company company company company company company company company company company company company company company company company company company company company company company company company company company company company company company company company company company company company company company company company company company company company company company company company company company company company company company company company company company company company company company company company company company company company company company company company company company company company company company company company company company company company company company company company company company company company company company company company company company company company company company company company company company.\nThe company and the\nThe company\nThe company\nThe\nThe The\n\n\n\n\n\n\n\n\n\n\n\n\n\n\n\n\n\n\n\n\n\n\nProducts\n\n\n\n\n\n\n\n\n\n\n\n\n\n The company is a leading provider of innovative, high-quality, and cost-effective solutions for the global businesses. The company is a leading provider of innovative, high-quality, and cost-effective'

Table 3: Responses for repeated tokens inputs, before and after patching

# D. Proofs

In this section we provide the omitted proofs from the main text. We start by clarifying our notation. Let $\mathbf{v}_i^{(\ell)}$, $\mathbf{q}_i^{(\ell)}$, and $\mathbf{k}_i^{(\ell)}$ be the $d$-dimensional value, query, and keys vectors of the $i$-th token at the $\ell$-th layer. Let $S_n = \left( \mathbf{v}_1^{(0)}, \ldots, \mathbf{v}_k^{(0)}, \underbrace{\mathbf{v}^{(0)}, \ldots, \mathbf{v}^{(0)}}_{n \text{ times}} \right)$ be a sequence of length $n + k$ composed of $k$ 'prefix' $\mathbf{v}_1^{(0)}, \ldots, \mathbf{v}_k^{(0)}$ tokens and $n$ copies of the token $\mathbf{v}^{(0)}$. Instead, let $S^* = \left( \mathbf{v}^{(0)} \right)$ be the singleton list consisting of the same token being repeated in the sequence $S_n$. For example, let $S_n = ('Hello', 'how', 'are', 'you', \ldots, 'you')$ where by, for example, 'Hello' we actually mean the $d$-dimensional vector embedding of the word 'Hello'. Then, the tokens 'Hello', 'how, 'are' would be the prefix tokens (i.e. $k = 3$) and we would have $S^* = ('you')$, i.e. 'you' would be the repeated token.

We study the same model architecture to the one studied by Barbero et al. (2025). We note that the exact architecture choice

is unlikely to affect the final conclusion of our results as we mainly rely on very general properties of Lipschitz continuity of the various components. In our proofs, we rely on the fact that attention *disperses* as the underlying sequence gets longer (Veličković et al., 2024). In particular, we use the fact that we can bound $\alpha_{i,j}^{\ell} \leq 1/n \exp(\delta)$ for some $\delta$ that is a function of the spectral norm of the weights entering the softmax, as shown by Veličković et al. (2024).

A Transformer block updates the value $\mathbf{v}^{(\ell)i}$ of token $i$ at layer $\ell$ as:

$$\mathbf{z}_i^{(\ell)} = \sum_{j \leq i} \alpha_{i,j}^{(\ell)} \mathbf{v}_j^{(\ell)} + \mathbf{v}_i^{(\ell)},$$

$$\mathbf{v}_i^{(\ell+1)} = \psi\left(\mathbf{z}_i^{(\ell)}\right) + \mathbf{z}_i^{(\ell)}$$

with $\alpha_{i,j}^{\ell}$ the attention coefficient between $i$ and $j$ of layer $\ell$ and $\psi$ an MLP. We write by $T^{(\ell)}$ the $\ell$-th Transformer layer, that takes as input a sequence $S$ and outputs a sequence $T^{(\ell)}(S)$ such that each element in the sequence is updated as above. We denote by $T^{(\ell)}(S)_i$ the vector corresponding to the $i$-th sequence element after the Transformer is applied. In our proofs we assume that $\psi$ is Lipschitz with constant $\|\nabla\psi\|$. This is immediate as the domain of $\psi$ is bounded and $\psi$ is assumed to be continuous.

**Lemma D.1.** *Consider the sequences $S_n$ and $S^*$ and a one-layer Transformer $T^{(1)}$. Then the representation of the final token of $S_n$ converges strongly to the representation of the single token of $S^*$, i.e.,*

$$\lim_{n \to \infty} \left\|T^{(1)}(S_n)_{n+k} - T^{(1)}(S^*)_1\right\| = 0. \tag{4}$$

*Proof.* It is required to show that there exists $N$ such that for all $n > N$ we have that $\|T(S_n)_{n+k} - T(S^*)_1\| < \epsilon$ for all $\epsilon > 0$. We use the simple estimate that $\alpha_{i,j}^{(\ell)} \leq 1/n \exp(\delta)$ with $n$ the sequence length and $\delta$ some constant that takes into account the largest difference in the activations (see Veličković et al. (2024)). We also let $r = \max_j \|\mathbf{v_j}\|$, which always exists due to the finite vocabulary and continuity of Transformer operations (again, see Veličković et al. (2024) for further details).

As the Transformer only has one layer, we drop layer superscripts for clarity. We start by showing that it suffices to show that $\left\|\mathbf{z}_{n+k}^* - \mathbf{z}_1\right\| < \epsilon$ as this implies:

$$\begin{aligned}
\|T(S_n)_{n+k} - T(S^*)_1\| &= \|\psi\left(\mathbf{z}_{n+k}\right) + \mathbf{z}_{n+k} - \psi\left(\mathbf{z}_1^*\right) - \mathbf{z}_1^*\| \\
&\leq \|\psi\left(\mathbf{z}_{n+k}\right) - \psi\left(\mathbf{z}_1^*\right)\| + \|\mathbf{z}_{n+k} - \mathbf{z}_1^*\| \\
&< \left(\|\nabla\psi\| + 1\right)\epsilon.
\end{aligned}$$

Since $\|\nabla\psi\|$ is constant this gives us what is desired. We now show that $\left\|\mathbf{z}_{n+k}^* - \mathbf{z}_1\right\| < \epsilon$:

$$\begin{aligned}
\left\| \mathbf{z}_{n+k}^{*} - \mathbf{z}_1 \right\| &= \left\| \left( \sum_{j \leq n+k} \alpha_{n+k,j} \mathbf{v}_j + \mathbf{v} \right) - (2\mathbf{v}) \right\| \\
&= \left\| \sum_{j \leq n+k} \alpha_{n+k,j} \mathbf{v}_j - \mathbf{v} \right\| \\
&= \left\| \sum_{j \leq k} \alpha_{n+k,j} \mathbf{v}_j + \sum_{k < j \leq n+k} \alpha_{n+k,j} \mathbf{v} - \mathbf{v} \right\| \\
&= \left\| \sum_{j \leq k} \alpha_{n+k,j} \mathbf{v}_j + \left( 1 - \sum_{j \leq k} \alpha_{n+k,j} \right) \mathbf{v} - \mathbf{v} \right\| \\
&= \left\| \sum_{j \leq k} \alpha_{n+k,j} \mathbf{v}_j - \left( \sum_{j \leq k} \alpha_{n+k,j} \right) \mathbf{v} \right\| \\
&\leq \sum_{j \leq k} \alpha_{n+k,j} \left\| \mathbf{v}_j \right\| + \left( \sum_{j \leq k} \alpha_{n+k,j} \right) \left\| \mathbf{v} \right\| \\
&\leq \frac{2rk \exp(\delta)}{n}.
\end{aligned}$$

As $r, k, \delta$ are all constants, then we have that $\left\| T^{(1)}(S_n)_{n+k} - T^{(1)}(S^*)_1 \right\|$ decays as $O(1/n)$ and is thus below any $\epsilon$ for large enough $n$. $\qquad\square$

Next, we extend the above result to a Transformer with $L$ layers, i.e. $T_L = T^{(L)} \circ T^{(L-1)} \circ \cdots \circ T^{(1)}$, where each $T^{(\ell)}$ denotes the mapping performed at layer $\ell$. Note that we use the superscript $T^{(\ell)}$ to denote the $\ell$-th Transformer layer, while the subscript $T_\ell$ to denote the composition of the first $\ell$ Transformer layers. We assume that each layer is Lipschitz continuous with respect to its input, with Lipschitz constant $\left\| \nabla T^{(\ell)} \right\|$. One can always find a Lipschitz constant as Transformers are continuous and, in our case, have a compact domain due to the finite vocabulary size.

**Theorem D.2.** *Let $T_L$ be a Transformer with $L$ layers and suppose that each layer is Lipschitz continuous with constant $\left\| \nabla T^{(\ell)} \right\|$. Then for the sequences $S_n$ and $S^*$,*

$$\lim_{n \to \infty} \left\| T^{(L)}(S_n)_{n+k} - T^{(L)}(S^*)_1 \right\| = 0. \tag{5}$$

*Proof.* We prove the result by induction on the number of layers $L$. The base case follows from Lemma D.1. Now assume that the result holds for a Transformer with $\ell$ layers, that is,

$$\left\| T_\ell(S_n)_{n+k} - T_\ell(S^*)_1 \right\| < \epsilon_n,$$

with $\epsilon_n \to 0$ as $n \to \infty$. Consider an $(\ell+1)$-layer Transformer $T_{\ell+1} = T^{(\ell+1)} \circ T_\ell$. Since $T^{(\ell+1)}$ is Lipschitz continuous with constant $\left\| \nabla T^{(\ell+1)} \right\|$, we have

$$\begin{aligned}
\left\| T_{\ell+1}(S_n)_{n+k} - T_{\ell+1}(S^*)_1 \right\| &= \left\| T^{(\ell+1)}\big(T_\ell(S_n)\big)_{n+k} - T^{(\ell+1)}\big(T_\ell(S^*)\big)_1 \right\| \\
&\leq \left\| \nabla T^{(\ell+1)} \right\| \left\| T_\ell(S_n)_{n+k} - T_\ell(S^*)_1 \right\| \\
&\leq \left\| \nabla T^{(\ell+1)} \right\| \epsilon_n.
\end{aligned}$$

Since $\epsilon_n \to 0$ as $n \to \infty$, it follows that

$$\lim_{n \to \infty} \left\| T_{\ell+1}(S_n)_{n+k} - T_{\ell+1}(S^*)_1 \right\| = 0.$$

We note as a technicality above that we tacitly ignore that we really consider the output of the model after applying a projection $\pi$ that only considers the projection to the last token. This detail can be fortunately ignored because the projection mapping $\pi$ has Lipschitz constant 1.

Thus, by induction, the result holds for any finite number of layers $L$. $\qquad\square$

