# OpenReview forum: "Interpreting the Repeated Token Phenomenon in Large Language Models"
_ICML.cc/2025/Conference — ICML 2025 poster_

### Official Review · Reviewer_xK8R · 2025-02-28

**Overall Recommendation:** 3

**Summary:**

This paper attempts the explain the phenomenon of "repeated tokens" with the behaviours of "attention-sink" in LLMs. Firstly, the authors find that besides the first token, the repeated tokens also have high attention scores and large hidden states norms. Then, the authors identify the neurons that contribute to the high norms. The authors explain this phenomenon by Equation 4 (which I think is the most important one): as the number of repeated tokens increases, the influence of prefix (which may related to the system instructions or user instructions) diminishes. Finally, the authors explore how to utilise the explanation to understand repeated token attack and mitigate the attack.


## update after rebuttal

I think most of my concerns have been addressed, so I raised my rating to 3. However, I understand that further investigation may be required here, which prevents me rating higher.

**Claims And Evidence:**

I think most of claims in this paper are supported by the empirical studies.

Regarding the attack mitigation, the authors propose a manual patch approach. In the appendix, they show that this editing method mitigates the attack by using several examples. I think more empirical studies may need to be conducted with some attack success metrics to further support the conclusion.

**Essential References Not Discussed:**

I think this paper has cited essential references.

**Experimental Designs Or Analyses:**

I think most of the experimental designs and analyses are sound and valid. However, I feel there is a gap between "repeated tokens could diminish the influence of the preceding prefix" and "repeated tokens could lead to training data leakage". I am looking forward to the clarifications from the authors.

**Methods And Evaluation Criteria:**

The authors first showcase the relationship between repeated tokens and attention sink with several case studies. Besides LLaMA2-7B model used in the main paper, the results for other models are also showcased in appendix.


Then based on the explorations above, the authors propose a mitigation approach. Although the authors demonstrate that the mitigation approach won't harm the model performance on unrelated task, they fail to show the effectiveness of mitigation approach towards repeated token attack using systematic benchmarks or evaluations. Only two cases are shown in the appendix.

**Other Comments Or Suggestions:**

See the above.

**Other Strengths And Weaknesses:**

I have included most of the strengths and weaknesses in the above.


Another weaknesses is that most of experiments are conducted using case studies. A systematic evaluation with metrics is lacking. Besides the attack success rate of mitigation approach I mentioned before, I find that when building the relationships between repeated tokens and attention sink, only some samples are provided. Why not include some statistical results to show the generalisation of this phenomenon?

**Questions For Authors:**

These questions are less related to my current ratings.

1. Do you have any intuitions on why repeated tokens induce model memorization?

2. Do you have any intuitions on why certain repetition cannot lead to attention sinks / large norm of activations? The paper about massive activations [1] also show that besides the first token, several other tokens with low semantics could also induce the attention sink. Do you think this is related?

3. I am aware that theoretical analysis of equation 4 is intrinsically difficult, do you have any theoretical intuitions?

[1] Sun et al. Massive Activations in Large Language Models. COLM 2024.

**Relation To Broader Scientific Literature:**

This paper is related to both mechanistic interpretability and safety in LLMs. I think it contributes to understanding the circuits of Transformer and why repeated tokens could attack LLMs. The mitigation approach proposed in this paper, if justified with more evidence, I think it will also contribute to the community of memorization attack.

**Theoretical Claims:**

Yes. Only claim 4.1 is a theoretical claim. The proof shown in this paper is correct.

---

> ### Author Rebuttal · Authors · 2025-03-30
>
> We thank the reviewer for their valuable comments. Next, we address the reviewer's questions and concerns:
>
> __“I feel there is a gap between "repeated tokens could diminish the influence of the preceding prefix" and "repeated tokens could lead to training data leakage". I am looking forward to the clarifications from the authors.__”
>
> We provide a brief proof below to theoretically validate the diminishing prefix influence (a full proof will be in the Camera-Ready version). This effect leads to the described attention sink divergence.
>
> We have revised the paper to include a theoretical analysis of equation 4. In short, the output of an attention head ($o$) for a sequence with a prefix (of size $k$) proceeded by a set of repeated tokens (of length $n$) is:
>
> $ o = \sum_{i<k} \alpha_i v_i  + \sum_{k <= i < n+k} \alpha _i v $
>
> where $\alpha_i$ are the coefficients of the attention matrix (after softmax), thus:
> $ \sum_{i}\alpha_i = 1 $
>
> $v_i$ is the value of token at index $i$, and $v$ is the value of the repeated token (from index $k$ to $n+k$)
>
> RoPE affects key and queries, thus affecting the values $a_i$ but the values of repeats of the same token are fixed $v$.
>
> For a long enough sequence the values of the repeated token will dominate the values of the prefix, this output will converge to $v$.
>
> While we acknowledge the need for further investigation, our work establishes a crucial _precursor_ to training data extraction. The identified mechanism of repeated token divergence leading to attention sinks creates the _conditions_ under which the model is more likely to retrieve and potentially output memorized content.
>
> __“Do you have any intuitions on why certain repetition cannot lead to attention sinks / large norm of activations? The paper about massive activations [1] also show that besides the first token, several other tokens with low semantics could also induce the attention sink. Do you think this is related?__”
>
> We suspect certain repetitions does not lead to attention sinks for several reasons:
> 1) the context window is too short for the convergence to take place.
> 2) the circuit identifying the first token does not generalize to all models.
>
> We suspect the model doesn't learn to identify <bos> but rather the first-token due to "packing", and the face that multiple <bos> tokens appeared in each context window, and believe that knowledge about the subtle training decisions will shed light on repetitions that do not lead to attention sinks.
>
> We think that BoS is a token with no-entropy, since it is always the first token, it does not attend the rest of the sequence, it is updated in a fixed manner in each layer. The model leverages that to control and bias the attention function. We suspect that other tokens, especially those carrying little information about the context (conjunctions, such as: "the", "and", "but"), will be used by some attention heads to implement a no-op. This has been studied in [1].
>
> [1] Outlier-Efficient Hopfield Layers for Large Transformer-Based Models, arxiv, 2024.
>
> __“Another weaknesses is that most of experiments are conducted using case studies. A systematic evaluation with metrics is lacking. Besides the attack success rate of mitigation approach I mentioned before, I find that when building the relationships between repeated tokens and attention sink, only some samples are provided. Why not include some statistical results to show the generalisation of this phenomenon?”__
>
> The causal relationship between the attention sink mechanism and the repeated token phenomenon is substantiated by our ablation experiments targeting specific 'sink neurons' (Section 3.2). Figure 3 provides empirical evidence that removing the contribution of these identified neurons effectively mitigates the high activation norms characteristic of repeated token divergence. This experimental approach, focusing on causal interventions within the identified circuit, is central to mechanistic interpretability and provides sufficient evidence to confirm the direct involvement of the attention sink mechanism in the observed phenomenon

---

> > ### Comment · Reviewer_xK8R · 2025-04-02
> >
> > Thank you for your response. I think most of my concerns have been addressed, and I understand that further investigation may be required here. I have updated my rating.
> >
> > I have an assumption here: if the repeated tokens could diminish the influence of the preceding tokens, such as system prompt / user prompt, then the distribution of generated response is closer to the data distribution in the pre-training. What is your opinion about this, or what is your hypothesis on why "repeated tokens could diminish the influence of the preceding prefix" leads to "repeated tokens could lead to training data leakage"?

---

> > > ### Author Response · Authors · 2025-04-02
> > >
> > > We appreciate the reviewer's acknowledgement of our rebuttal.
> > >
> > > We also recognize the valuable point raised. Our current hypothesis suggests that reducing the influence of the system prompt (prefix) is a significant factor in circumventing model alignment. We further suspect this manipulation, along with interference in the Attention mechanism, contributes to the leakage of training data. A dedicated study exploring the causal link between these two factors (prefix diminution and Attention disruption) and training data exposure remains an area for future investigation.

---

### Official Review · Reviewer_49D9 · 2025-03-11

**Overall Recommendation:** 4

**Summary:**

This paper studies the "repeated token divergence" phenomenon in LLMs, where models fail to accurately repeat a single token when instructed to do so. The authors provide a mechanistic explanation linking this behavior to attention-sinks (where the initial token in a sequence receives disproportionately high attention).
The researchers identify a specific neural circuit responsible for attention-sinks consisting of two key stages:
1) the first attention layer "marks" the initial token;
2) specific MLP neurons (termed "sink neurons") add high-magnitude values to its hidden state, creating the attention-sink.

When processing sequences with repeated tokens, the model's first attention layer confuses these repetitions with the beginning-of-sequence (BoS) token, triggering the same neural circuit and causing the model to diverge from its intended output, sometimes revealing memorized training data.
They develop and validate a targeted patch that corrects the issue without significantly impacting model performance on standard benchmarks.

**Claims And Evidence:**

The cluster attack's effectiveness is asserted but not quantitatively measured, making it difficult to assess its practical significance compared to direct token repetition.

**Essential References Not Discussed:**

N/A

**Experimental Designs Or Analyses:**

- The paper doesn't include control experiments with non-identical but similar tokens to test the specificity of the mechanism.

- The validation of the patch focuses on model performance on standard tasks but doesn't directly measure its effectiveness at preventing the extraction of training data, which is the primary security concern.

**Methods And Evaluation Criteria:**

The methodological approach is generally sound, but the paper lacks a systematic evaluation of how many tokens from the vocabulary can induce attention sinks when repeated, instead showing selected examples.

**Other Comments Or Suggestions:**

N/A

**Other Strengths And Weaknesses:**

other strenghts:

- The authors use a systematic approach to identify the neural mechanisms involved, using causal interventions (neuron ablation) to verify their hypotheses. This provides strong evidence for their claims about the role of specific neurons in creating attention-sinks.

other weaknesses:

- The paper shows that not all tokens can effectively induce attention-sinks when repeated (Figure 5). This unexplained variability suggests additional factors at play that aren't fully captured by the current mechanistic account.
- While the paper identifies the mechanism behind repeated token divergence, it doesn't fully explain why this leads to training data extraction. The relationship between attention-sinks and the retrieval of memorized content remains unclear.
- The authors note that while there are shared motifs across LLMs, the first attention layer's specific behavior was unique to LLaMa2. This raises questions about how broadly applicable their exact mechanisms are across different model architectures.

**Questions For Authors:**

N/A

**Relation To Broader Scientific Literature:**

The paper presents convincing evidence for its core mechanistic explanation of repeated token divergence through attention sinks. The empirical work identifying specific neural circuits is particularly strong. However, the connection between this mechanism and training data extraction needs more thorough study.

**Theoretical Claims:**

In Claim 4.1 and its proof:
the proof makes logical sense but relies on simplifying assumptions. It correctly identifies that with identical tokens, the value vectors in self-attention are equal. However, the proof doesn't fully account for the positional information encoded by RoPE: while RoPE affects keys and queries, the interaction between position-encoded representations could theoretically still allow differentiation between positions.
The empirical observation in Equation 4 helps support this claim, but the theoretical argument could be more rigorous in addressing how RoPE specifically fails to distinguish positions in this context.

---

> ### Author Rebuttal · Authors · 2025-03-30
>
> We thank the reviewer for their valuable comments. Next, we address the reviewer's questions and concerns:
>
> __“The paper shows that not all tokens can effectively induce attention-sinks when repeated (Figure 5). This unexplained variability suggests additional factors at play that aren't fully captured by the current mechanistic account.__”
>
> As we mentioned in the limitations section, some additional factors can indeed be at play. Nevertheless, our goal is to understand why and how the previously identified phenomenon of model divergence due to repeated tokens, happens. Our current model is _useful_ - it allows intervention that prevents the phenomenon, and therefore we believe that our analysis is valuable.
>
> __“While the paper identifies the mechanism behind repeated token divergence, it doesn't fully explain why this leads to training data extraction. The relationship between attention-sinks and the retrieval of memorized content remains unclear.__”
>
> While we acknowledge the need for further investigation, our work establishes a crucial _precursor_ to training data extraction. The identified mechanism of repeated token divergence leading to attention sinks creates the _conditions_ under which the model is more likely to retrieve and potentially output memorized content.
>
> __“The authors note that while there are shared motifs across LLMs, the first attention layer's specific behavior was unique to LLaMa2. This raises questions about how broadly applicable their exact mechanisms are across different model architectures.__”
>
> We suspect subtle differences in training, data or initialization induce different ways for the first attention layer to behave. However, it was important to us to show that by reverse-engineering one layer we can generate non-repeating sequences which also induce attention sinks.
>
> __“The validation of the patch focuses on model performance on standard tasks but doesn't directly measure its effectiveness in preventing the extraction of training data, which is the primary security concern.__”
>
> We have provided a few examples showing how coherency is maintained. It is rather difficult to measure training data leakage as most models are open-weights but not fully open-source, therefore there is very little information about their training data.
>
> __“In Claim 4.1 and its proof: the proof makes logical sense but relies on simplifying assumptions. It correctly identifies that with identical tokens, the value vectors in self-attention are equal. However, the proof doesn't fully account for the positional information encoded by RoPE: while RoPE affects keys and queries, the interaction between position-encoded representations could theoretically still allow differentiation between positions. The empirical observation in Equation 4 helps support this claim, but the theoretical argument could be more rigorous in addressing how RoPE specifically fails to distinguish positions in this context.”__
>
> We have revised the paper to include a theoretical analysis of equation 4. We argue that differentiation between positions of equivalent tokens is _impossible_ with RoPE. In short, the output of an attention head ($o$) for a sequence with a prefix (of size $k$) proceeded by a set of repeated tokens (of length $n$) is:
>
> $ o = \sum_{i<k} \alpha_i v_i  + \sum_{k <= i < n+k} \alpha _i v $
>
> where $\alpha_i$ are the coefficients of the attention matrix (after softmax), thus:
> $ \sum_{i}\alpha_i = 1 $
>
> $v_i$ is the value of token at index $i$, and $v$ is the value of the repeated token (from index $k$ to $n+k$)
>
> RoPE affects key and queries, thus affecting the values $a_i$ but the values of repeats of the same token are fixed $v$.
>
> For a long enough sequence the values of the repeated token will dominate the values of the prefix, this output will converge to $v$.

---

> > ### Comment · Reviewer_49D9 · 2025-04-05
> >
> > I thank the authors for providing these clarifications, and i'd like to keep the overall score to 4.

---

### Official Review · Reviewer_LKw5 · 2025-03-13

**Overall Recommendation:** 3

**Summary:**

- This paper discusses the repeated token divergence issue, links it to a possible LLM phenomenon named attention sink and proposes a solution for it.
 - This paper takes a Mechanistic Interpretability approach, analyzing the underlying mechanism of attention sinks in LLMs, and shows empirical evidence of this mechanism. Then the analysis is linked with the repeated token divergence issue.
 - The authors introduce a "cluster attack" based on their findings, which extends the vulnerability beyond simple token repetitions. They also develop a targeted patch that mitigates the issue without harming the model's performance on other tasks.

**Claims And Evidence:**

The claims are proved to some extent but not fully convincing.
- line 294: Rope affects the queries and keys, not keys and values,  this claim: "thus symmetry between all tokens is preserved"  shall be further clarified.
- line 299: Rope will affects attention output according to the relative positions, so even the repeated tokens will have different outputs, the claim: "all tokens in the repeated sequence will have the same output after the first attention layer." in line 299 is confusing.

**Essential References Not Discussed:**

None

**Experimental Designs Or Analyses:**

Although mentioned in the limitations, the first attention  layer’s behavior was unique to LLaMa2,  but the first attention layer mechanism poses an important finding in this paper, which hinders the generalization ability of this paper. How well can this method be used in other more models is unknown.

**Methods And Evaluation Criteria:**

The methods and evaluation criteria make sense.

**Other Comments Or Suggestions:**

None

**Other Strengths And Weaknesses:**

weakness:
- The reason behind the high attention scores of the attention sinks and repeated tokens seems different. Repeated tokens are likely to have high activations because of self-attention between similar embeddings, but attention sinks are due to other different reasons. In addition, does the LLm distinguish attention sinks only by attention scores? Why can't model distinguish by positions?
- The line 294, Rope affects the queries and keys, not keys and values,  this claim: "thus symmetry between all tokens is preserved"  shall be further clarified. In addition, Rope will affects attention output according to the relative positions, so even the repeated tokens will have different outputs, the claim: "all tokens in the repeated sequence will have the same output after the first attention layer." in line 299 is confusing.

**Questions For Authors:**

- Although mentioned in the limitations, the first attention  layer’s behavior was unique to LLaMa2,  but the first attention layer mechanism poses an important finding in this paper, which hinders the generalization ability of this paper. How well can this method be used in other more models is unknown.
- I wonder how will the mitigation influence the decoding efficiency as all sink neurons need to be figured out.

**Relation To Broader Scientific Literature:**

This paper are related prior literatures in terms of several aspects:
- repeated token divergence
- Attention sink in transformer network
- LLM attack

**Theoretical Claims:**

Yes, there is not much theoretical claims in this paper, most results are obtained empirically.

---

> ### Author Rebuttal · Authors · 2025-03-30
>
> We thank the reviewer for their valuable comments. Next, we address the reviewer's questions and concerns:
>
> __“Rope affects the queries and keys, not keys and values, this claim: "thus symmetry between all tokens is preserved" shall be further clarified. Rope will affect attention output according to the relative positions, so even the repeated tokens will have different outputs, the claim: "all tokens in the repeated sequence will have the same output after the first attention layer." in line 299 is confusing.”__
>
> We agree and sorry for the confusion. We have revised the paper to include a proof to these claims. In short, the output of an attention head ($o$) for a sequence with a prefix (of size $k$) proceeded by a set of repeated tokens (of length $n$) is:
>
> $ o = \sum_{i<k} \alpha_i v_i  + \sum_{k <= i < n+k} \alpha _i v $
>
> where $\alpha_i$ are the coefficients of the attention matrix (after softmax), thus:
> $ \sum_{i}\alpha_i = 1 $
>
> $v_i$ is the value of token at index $i$, and $v$ is the value of the repeated token (from index $k$ to $n+k$)
>
> RoPE affects key and queries, thus affecting the values $a_i$ but the values of repeats of the same token are fixed $v$.
>
> For a long enough sequence the values of the repeated token will dominate the values of the prefix, this output will converge to $v$.
>
> __“Although mentioned in the limitations, the first attention layer’s behavior was unique to LLaMa2, but the first attention layer mechanism poses an important finding in this paper, which hinders the generalization ability of this paper. How well this method can be used in other models is unknown.”__
>
> We suspect subtle differences in training, data or initialization induce different ways for the first attention layer to behave. However, it was important to us to show that by reverse-engineering one layer we can generate non-repeating sequences which also induce attention sinks.
>
> __“I wonder how will the mitigation influence the decoding efficiency as all sink neurons need to be figured out.”__
>
> The decoding efficiency after the mitigation is not significantly different from the original decoding efficiency. As can be seen by the pseudo-code (Listing 1), our patch changes one neuron for LLaMa2. Even with more neurons, the patch can be done in parallel on all of them as the set of neurons can be precomputed.
>
> __“The reason behind the high attention scores of the attention sinks and repeated tokens seems different. Repeated tokens are likely to have high activations because of self-attention between similar embeddings, but attention sinks are due to other different reasons. In addition, does the LLm distinguish attention sinks only by attention scores? Why can't model distinguish by positions?”__
>
> Our analysis indicates that the high attention scores observed in both phenomena are not due to distinct causes but rather stem from the same underlying neural mechanism. Specifically, the mechanism responsible for creating attention sinks (high attention on the initial token, crucial for fluency ) is inadvertently triggered by long sequences of repeated tokens.
>
> The core issue lies in the model's first attention layer, which struggles to differentiate a sequence of identical tokens from a single, initial token, particularly in architectures using RoPE. This layer misidentifies the repeated sequence, activating the same "sink neurons" that amplify the hidden states of initial tokens. Consequently, repeated tokens acquire high hidden-state norms, attracting disproportionate attention similar to the BoS token. While the model utilizes positional information (e.g., via causal masking ), this specific case of identical repetitions challenges its ability to distinguish position effectively, leading to the observed divergence.
>
> Therefore, our work demonstrates that repeated token divergence is mechanistically linked to the attention sink phenomenon, arising from a failure in positional differentiation under specific repetitive conditions. We hope this clarifies the connection established in our findings.

---

> > ### Comment · Reviewer_LKw5 · 2025-04-07
> >
> > Thank you for the rebuttal to clarify my questions, I will update my score accordingly.

---

### Official Review · Reviewer_TFmf · 2025-03-14

**Overall Recommendation:** 3

**Summary:**

This paper focuses on the repeat phenomenon in LLMs. The authors view this problem from the view of attention sink. They show that the first attention layer marks the initial token, and the later MLP neuron amplifies its hidden state, creating an attention sink. A patching method is proposed to mitigate this effect.

**Claims And Evidence:**

The authors claim that

1. The repeated token divergence is caused by attention sinks.

2. Repeated tokens activate the same neural circuit responsible for attention sinks.

3. A simple patching can mitigate the issue.

These claims are supported by the evidence.

**Essential References Not Discussed:**

N.A.

**Experimental Designs Or Analyses:**

The experimental designs in the paper are plausible.

**Methods And Evaluation Criteria:**

The authors propose the patching method to mitigate the sink issue. The experiments demonstrate the efficiency of the proposed method.

**Other Comments Or Suggestions:**

N.A.

**Other Strengths And Weaknesses:**

The main weakness of this work is that the task considered is very rare in the realistic application. Most users of LLMs will not let LLMs repeat some words. The lack of motivation discounts the importance of this work.

**Questions For Authors:**

The main concern is the importance of the repeat task. It is hard to justify why people want the LLMs to repeat tokens.

**Relation To Broader Scientific Literature:**

N.A.

**Theoretical Claims:**

No theoretical analysis is provided in the paper.

---

> ### Author Rebuttal · Authors · 2025-03-29
>
> We thank the reviewer for their valuable comments. Next, we address the reviewer's questions and concerns:
>
> __“The main weakness of this work is that the task considered is very rare in the realistic application. Most users of LLMs will not let LLMs repeat some words. The lack of motivation discounts the importance of this work. It is hard to justify why people want the LLMs to repeat the token.”__
>
> The repeating tokens “task” is indeed not a common task. Nevertheless, once this behavior is triggered, it can cause a training data leakage – a serious security flow [1, 2]. Like many software vulnerabilities, the behavior that triggers the presented vulnerability is not a typical user input. Nevertheless, we assume here an adversarial user that aims to attack the model, and we find a patch that can protect the model from such an attacker.
>
>
> [1] Nasr et al. Scalable extraction of training data from (production) language models, ICLR 2025
>
> [2] Bye Bye Bye...: Evolution of repeated token attacks on ChatGPT models, https://dropbox.tech/machine-learning, 2024

---

### Decision · Program_Chairs · 2025-05-01

**Decision:**

Accept (poster)

**Comment:**

This paper explains the mechanism behind LLMs' failure to accurately repeat a single word multiple times by linking it to the "attention-sink" phenomenon. The authors identify a specific neural circuit where the first attention layer marks the initial token and a later MLP neuron adds high-magnitude values to its hidden state, creating the sink. This circuit is disrupted when processing long repetitions, causing the model to drift from instructions, sometimes leaking training data. All reviewers recommended acceptance, appreciating the paper's mechanistic interpretability approach, strong empirical evidence, and the proposed targeted patch that fixes the issue without harming model performance. During the rebuttal, the authors effectively addressed concerns about the theoretical foundations of their work, generalizability to other models, and the connection between attention sinks and training data leakage. Given the paper's novel contributions to understanding and mitigating a security vulnerability in LLMs, I recommend accepting this paper for ICML 2025.